# OpenMathInstruct-2: Accelerating AI for Math with Massive Open-Source Instruction Data

**Shubham Toshniwal**
NVIDIA
stoshniwal@nvidia.com

**Wei Du**
NVIDIA
wedu@nvidia.com

**Ivan Moshkov**
NVIDIA
imoshkov@nvidia.com

**Branislav Kisacanin**
NVIDIA
Institute for AI R&D of Serbia
Faculty of Technical Sciences,
University of Novi Sad
bkisacanin@nvidia.com

**Alexan Ayrapetyan**
NVIDIA
aayrapetyan@nvidia.com

**Igor Gitman**
NVIDIA
igitman@nvidia.com

## Abstract

Mathematical reasoning continues to be a critical challenge in large language model (LLM) development, with a significant performance gap between closed-source and open-source efforts, largely due to differences in training data quality and scale. The emergence of *frontier* open-weight LLMs offers new opportunities to generate high-quality, commercially permissible synthetic data to help bridge this gap. In this paper, we investigate the recently released Llama3.1 family of models to improve open-source math reasoning through synthetically generated supervised finetuning (SFT) data. We conduct ablation studies to optimize design choices for the dataset, such as solution format and teacher model selection, which enhance SFT performance. We also investigate SFT's robustness to incorrect solutions and find that at large data scales, the model can be robust to as much as 20% noise, suggesting that the simple answer-matching heuristic is sufficient for SFT data selection. Based on these insights, we create the OpenMathInstruct-2 dataset which consists of 14M question-solution pairs ($>$ 600K unique questions), making it nearly eight times larger than any previous such dataset. Finetuning the `Llama-3.1-8B-Base` using OpenMathInstruct-2 outperforms `Llama3.1-8B-Instruct` on MATH by an absolute 14.6% ($51.9 \rightarrow 66.5$), demonstrating the effectiveness of the dataset. As part of our open-source efforts, we will release the code, the finetuned models, and the OpenMathInstruct-2 dataset under a commercially permissive license.[1]

## 1 Introduction

Synthetic data has emerged as a key technique for building large language models due to its cost-effectiveness and scalability [Meta-AI, 2024, NVIDIA, 2024, DeepSeek-AI, 2024b]. In particular, synthetic data is well suited for mathematical reasoning where the performance improvements with synthetic data scaling are yet to saturate [Zeng et al., 2024, Chan et al., 2024, Yang et al., 2024]. However, access to this progress is limited because the current largest math datasets remain *closed-source* [Zeng et al., 2024, Yang et al., 2024]. The closed nature of these datasets introduces two major issues. First, concerns over data leakage erode trust in reported benchmark results [Aiyappa

---

[1]Data and models are available at https://huggingface.co/collections/nvidia/openmath-2-66fb142317d86400783d2c7b
Code is available at https://github.com/Kipok/NeMo-Skills

MATH-AI@ NeurIPS 2024.

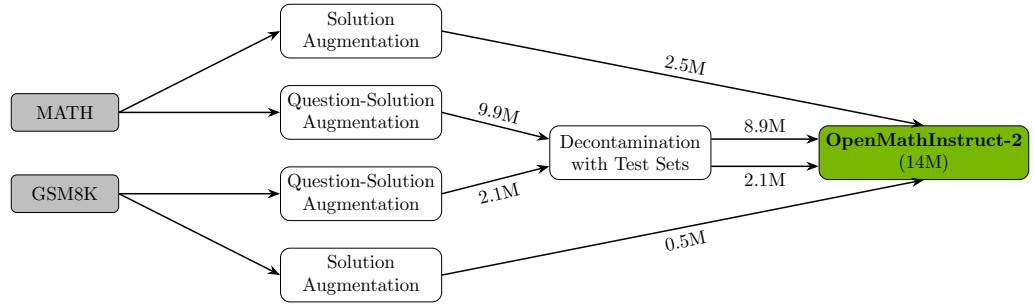

Figure 1: Overview of the OpenMathInstruct-2 construction pipeline.

et al., 2023]. For e.g., Zhang et al. [2024] show a drop of more than 10% for popular LLMs on an unpublished test set which is distributionally similar to the popular grade school math benchmark GSM8K [Cobbe et al., 2021]. Second, it prevents practitioners from fully understanding the impact of data composition and algorithmic choices [Soldaini et al., 2024].

Among open-source alternatives, the recent NuminaMath dataset [Li et al., 2024] has the largest collection of questions collected from diverse sources. However, its restrictive license—likely due to the use of GPT-4o in data processing and synthesis—limits its broader use. Similarly, other popular math instruction tuning datasets, such as MetaMathQA [Yu et al., 2024] and MathInstruct [Yue et al., 2024], have also relied on GPT models for data synthesis, which prohibits their usage in non-commercial settings. A notable exception is the OpenMathInstruct-1 [Toshniwal et al., 2024] dataset, one of the biggest open-source math reasoning datasets, where solutions are synthesized using open-weight models. However, OpenMathInstruct-1 has two key limitations. Firstly, its question diversity is constrained, since all the questions in the dataset are drawn from the training sets of MATH [Hendrycks et al., 2021] and GSM8K [Cobbe et al., 2021]. Secondly, at the time of its release, there was a sizable gap in the math reasoning capabilities of open and closed-source models. As a result, the dataset underrepresents more challenging problems compared to its GPT-based counterparts [Gou et al., 2024].

The emergence of *frontier* open-weight models [Meta-AI, 2024, DeepSeek-AI, 2024b] has made it possible to create high-quality, commercially permissible math reasoning datasets. In this paper, we use the recently released Llama3.1 family of models to generate synthetic math instruction tuning (SFT) data, and evaluate the quality of the math reasoning data by finetuning the Llama-3.1-8B-Base model. To create the dataset, we conduct careful ablation studies using the MATH dataset to determine design choices that impact the final SFT performance. The highlights of our findings include:

- *Chain-of-Thought (CoT) Solution Format*: Excessive verbosity can be detrimental to the SFT performance. Our proposed CoT format outperforms Llama's CoT format while being 40% shorter in solution length (see Figure 2 for a sample solution).

- *Choice of Teacher Model*: The SFT performance mirrors the teacher model's performance even when controlling for the SFT data size. Specifically, we find that finetuning `Llama3.1-8B-Base` on solutions generated by `Llama3.1-405B-Instruct` outperforms solutions generated by the model itself even when controlling for the SFT data size.

- *Robustness of SFT*: At sufficient data scale ($\geq$ 256K), the SFT performance suffers minimal to no degradation with as much as 20% incorrect solutions. Additionally, removing solutions with incorrect intermediate steps, as determined by LLM-as-a-Judge or the Nemotron Reward model [NVIDIA, 2024], also yields no performance benefit.

- *Impact of Question Diversity*: Controlling for SFT data size, we find that question diversity has a huge positive impact on SFT performance.

Based on the above findings, we create OpenMathInstruct-2 with data synthesized using Llama-3.1-405B-Instruct. The dataset uses the MATH and GSM8K training set questions and uses the LLM to (a) synthesize solutions to the original training set questions and (b) create new question-solution pairs similar to the training set questions. To ensure there is no test set contamination among the

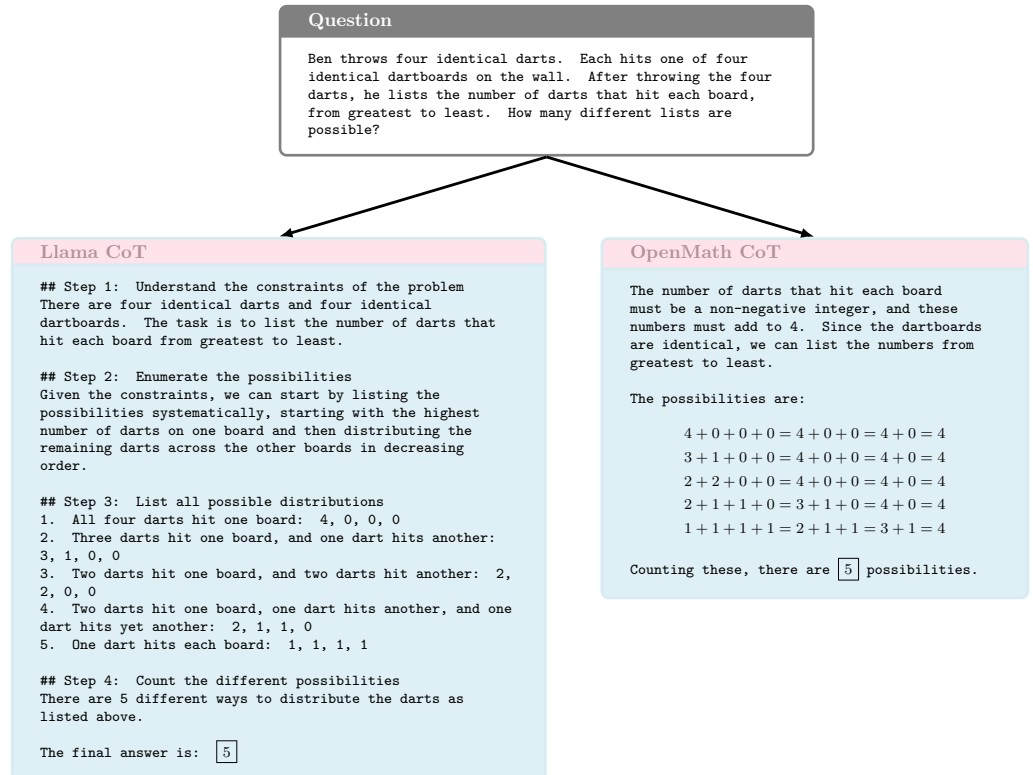

Figure 2: Comparing solutions in the Llama CoT format vs. the OpenMath CoT format for a sample question.

synthesized questions, we perform thorough decontamination using the `lm-sys` pipeline, followed by manual inspection [Yang et al., 2023]. Figure 1 provides an overview of the entire dataset construction pipeline. The final dataset consists of 14M question-solution pairs with 600K unique questions. Thus, OpenMathInstruct-2 is about 8 times bigger than the previous biggest standalone open-source dataset [Toshniwal et al., 2024].

The quality of OpenMathInstruct-2 is illustrated by the strong performance of the finetuned models. The `OpenMath2-Llama3.1-8B` model, which is the `Llama3.1-8B-Base` model finetuned with OpenMathInstruct-2, outperforms `Llama3.1-8B-Instruct` by an absolute 14.6% on MATH with just SFT. With a performance of 66.5 on MATH, `OpenMath2-Llama3.1-8B` is one of the strongest sub-10B open-source models. We will release all our fine-tuned models, code, the OpenMathInstruct-2 dataset, and a dataset explorer.

## 2 Experimental Setup

**Training Details.** For all the experiments, except when training on the full dataset, we train the `Llama3.1-8B-Base` model for four epochs and save a checkpoint at the end of every epoch. For the full dataset, we train the model for about 2.2 epochs (60K steps) and save six equally spaced checkpoints. The final checkpoint is created by averaging all the saved checkpoints. A global batch size of 512 is used along with the AdamW optimizer [Loshchilov and Hutter, 2019] with a learning rate of 5e-6 and a weight decay of 1e-2. All experiments are performed using the NeMo toolkit [Kuchaiev et al., 2019].

**Evaluation Setup.** We evaluate the final finetuned model on popular math reasoning benchmarks, namely GSM8K, MATH, AMC 2023, and AIME 2024. The finetuned model is evaluated in the zero-shot setting with greedy decoding.

Table 1: Comparison of our *OpenMath2-Llama* model with other sub-20B open-weight and open-source models without tool usage. Open-weight models finetuned with publicly released data are considered as open-source for the purposes of this table.

| | Model | GSM8K | MATH | AMC'23 | AIME'24 |
|---|---|---|---|---|---|
| Open Weight | DeepSeek-Coder-V2-Lite-Instruct [DeepSeek-AI, 2024a] | 86.4 | 61.8 | - | 0/30 |
| | Qwen2.5-Math-7B-Instruct [Yang et al., 2024] | 95.2 | 83.6 | 25/40 | 5/30 |
| | Llama3.1-8B-Instruct [Meta-AI, 2024] | 84.5 | 51.9 | 9/40 | 2/30 |
| Open Source | NuminaMath-7B-CoT [Li et al., 2024] | 75.4 | 55.2 | 11/40 | 0/30 |
| | OpenMath2-Llama3.1-8B (ours) | 92.7 | 66.5 | 16/40 | 2/30 |

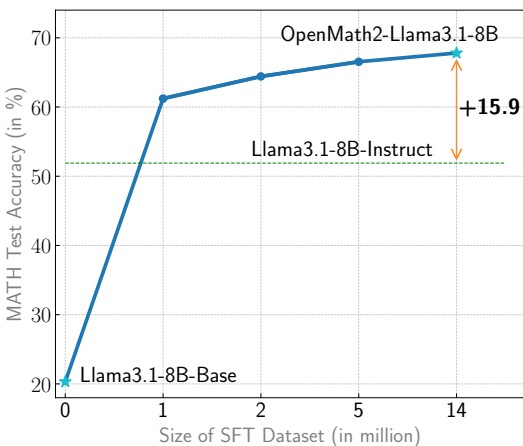

Figure 3: MATH Test Accuracy as a function of the SFT data size.

# 3 Results

To understand the impact of data scaling, we downsample the full dataset to 1M, 2M, and 5M-sized instruction tuning datasets using *fair downsampling* [Toshniwal et al., 2024]. Figure 3 plots the performance on the MATH test set with the increase in SFT data size. With even the 1M downsampled version of OpenMathInstruct-2, the final model easily outperforms `Llama3.1-8B-Instruct`. Finally, we observe a consistent gain with an increase in data size, and even at 14M dataset size, we see no signs of saturation in performance gains.

Table 1 presents the results for top-performing, sub-20B, open-weight and open-source models (without tool use). The `OpenMath2-Llama3.1-8B` model, which is finetuned on the full OpenMathInstruct-2 dataset, outperforms or matches `Llama3.1-8B-Instruct` on all the math reasoning benchmarks. Among the open-source models, we outperform the recently released `NuminaMath-7B-CoT` on all benchmarks as well. Finally, among all the presented models, the `OpenMath2-Llama3.1-8B` is second only to the `Qwen2.5-Math-7B-Instruct`, which has been trained on more than a trillion synthetically generated math reasoning tokens, and starts with a base model, `Qwen2.5-Math`, which is about 35% better than `Llama3.1-8B-Base`. [2]

# 4 Conclusion

We introduce OpenMathInstruct-2, a math instruction tuning dataset with 14M question-solution pairs and more than 600K unique questions. The dataset is created using `Llama3.1-405B-Instruct` model and released with a commercially permissive license. Compared to previous work, OpenMathInstruct-2 is about eight times larger than the previous biggest open-source dataset for

---

[2]We are unsure of the $n$-gram based data contamination protocol followed by `Qwen2.5-Math` given its obvious weakness in detecting paraphrases. In our own decontamination setup, which we borrow from Yang et al. [2023], we find paraphrases of test set questions that are identified by our pipeline but which $n$-gram matching will miss out on.

math reasoning. To support the open-source efforts, we will publicly release all the finetuned models, code, and the OpenMathInstruct-2 dataset.

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
