# OpenReview forum: "OpenMathInstruct-2: Accelerating AI for Math with Massive Open-Source Instruction Data"
_NeurIPS.cc/2024/Workshop/MATH-AI — MATH-AI 24_

### Official Review · Reviewer_NeAd · 2024-10-02

**Rating:** 7
**Confidence:** 4

**Review:**

The paper makes a significant contribution to the field of mathematical reasoning in LLMs by introducing a large, high-quality open-source dataset and demonstrating strong performance improvements. The work is thoroughly supported by extensive experiments, including the exploration of data scaling laws for supervised fine-tuning (SFT) and the robustness of SFT.

Overall, OpenMathInstruct-2 represents a major advancement, providing a massive dataset that can serve as a vital resource for open-source LLMs focused on mathematical reasoning. This significantly addresses the limitations of previous small datasets and enhances the potential for open-source AI development in this domain. However, I have the following two suggestions to further improve the manuscript of this paper:

1.	Limited Exploration of Dataset Quality Metrics: While the paper provides robust performance benchmarks, it could benefit from deeper insights into the qualitative aspects of the dataset. For example, analyzing the diversity of questions or the depth of reasoning required would offer a clearer understanding of the dataset’s strengths and potential gaps.

2.	Lack of Stopping Criterion for SFT: This paper does not offer guidance on when to stop fine-tuning in terms of performance gains versus computational costs, which would be valuable for practical applications.

---

### Official Review · Reviewer_qgb4 · 2024-10-07
**A good large-scale, open-source dataset designed to improve the performance of mathematical reasoning in large language models (LLMs)**

**Rating:** 8
**Confidence:** 4

**Review:**

The methodology is robust, employing the Llama3.1 family of models to generate high-quality synthetic data. The results demonstrate significant improvements and the experimental setup is thorough, with detailed training protocols and evaluation on multiple benchmarks such as MATH, GSM8K, AMC 2023, and AIME 2024.
The paper is well-structured and clearly written.
One area where the paper could improve is in providing more comprehensive comparisons to closed-source models, which would give a broader context for interpreting the performance gains.

Pros:
- Large scale dataset (14 mil)
- Significant performance improvement
- Contribution to the open source
- Well written paper

Cons:
- No comparison to the closed state of the art models
- Limited novelty in terms of groundbreaking new methods

Overall, OpenMathInstruct-2 is a commendable advancement. Addressing the minor limitations noted could further enhance its impact and utility. While the paper is not entirely without shortcomings, the strengths far outweigh the weaknesses.

---

### Official Review · Reviewer_4Dt3 · 2024-10-07
**This paper create 14M question-solution pairs math dataset for finetuning, and improve the open-source math reasoning Llama3-8B**

**Rating:** 6
**Confidence:** 4

**Review:**

I think this is a good paper for acceptance. only one question come in my mind.

Questions
- Aside from the dataset size, are there any other differences between OpenMathInstruct-1 and OpenMathInstruct-2?

Pros
- Data Generation with Llama 3-405B Model:-The dataset is generated using the Llama 3-405B model, making the process reproducible.
- Large-Scale Dataset:
- It contains more than 14M question-answer pairs, providing a comprehensive training foundation.

Cons:
See question.
Model Replacement Consideration:
- Could the Llama model be replaced with another similar model for data generation and fine-tuning?

---

### Official Review · Reviewer_Wgbg · 2024-10-07
**A new dataset which significantly improves LLM SFT**

**Rating:** 7
**Confidence:** 3

**Review:**

Overview: The necessity for high-quality datasets available for non-commercial research is undeniable. Access to robust datasets not only facilitates a deeper understanding of the domain but also propels forward the advancements within the field. This paper addresses this need effectively.

Strengths:
1. The idea of conducting an ablation study to pinpoint the crucial elements within the data that significantly influence LLM SFT is commendable. The insights derived from this study appear to be well-founded and logically presented.
2. The newly generated dataset has demonstrably enhanced performance metrics, which strongly supports the underlying hypothesis of the study. This advancement is a significant step forward in the application of LLM SFT.
3. The evaluation methodology is lucid and meticulous, effectively showcasing the proposed model's capabilities in comparison to established baselines. This clarity in presentation adds to the credibility of the results.
4. The introduction of the paper is thorough, setting a solid foundation by delineating the problem and summarizing the current landscape of related work. This context is invaluable for readers to appreciate the contributions of the paper.

Weaknesses:
1. Despite the promising results discussed in highlights 2, 3, and 4, there is a noticeable absence of rigorous evaluation studies to substantiate these claims. This gap might lead to skepticism regarding the replicability and validity of the reported improvements.
2. The paper does not provide a rationale for the specific size of the generated dataset—600K entries—nor does it explore the potential impact of scaling the dataset size. A discussion on how varying the dataset size might affect the performance would have been beneficial for understanding the scalability and limitations of the proposed approach.

Conclusion: Overall, the contributions of this paper are significant, particularly in the enhancement of training data for SFT applications. The introduction of a better-quality dataset not only enriches the available resources for researchers but also potentially sets a new benchmark in the field. I believe that the community will find great value in these contributions, despite the mentioned shortcomings.

---

### Decision · Program_Chairs · 2024-10-09

Accept